# An Evaluation of Large Language Models in Bioinformatics Research

## Abstract

Large language models (LLMs) such as ChatGPT have gained considerable interest across diverse research communities. Their notable ability for text completion and generation has inaugurated a novel paradigm for language-interfaced problem solving. However, the potential and efficacy of these models in bioinformatics remain incompletely explored. In this work, we study the performance LLMs on a wide spectrum of crucial bioinformatics tasks. These tasks include the identification of potential coding regions, extraction of named entities for genes and proteins, detection of antimicrobial and anti-cancer peptides, molecular optimization, and resolution of educational bioinformatics problems. Our findings indicate that, given appropriate prompts, LLMs like GPT variants can successfully handle most of these tasks. In addition, we provide a thorough analysis of their limitations in the context of complicated bioinformatics tasks. We envision this work to provide new perspectives and motivate future research in the field of both LLMs applications and bioinformatics.

## 1 Introduction

Large language models (LLMs) (Birhane et al., 2023; Katz et al., 2022; Li et al., 2022) such as GPT variants, which are neural network models trained on large amounts of unlabeled data, have recently attracted significant attention across a variety of research communities. Trained with a combination of unsupervised pre-training, supervised fine-tuning, and human feedback, LLMs can generate fluid and reasonable contextual conversations with text-based input queries, i.e. *prompts*. Such a natural language interface offers a versatile problem-solving platform, addressing tasks from text drafting to mathematical problem solving (Ouyang et al., 2022; Xu et al., 2023), which exceeds the capabilities of traditional single natural language processing models (Pang et al., 2002; Marrero et al., 2013; Lee et al., 2011). It is particularly noted that LLMs have demonstrated remarkable proficiency in human-like language generation and exhibited a discernible level of reasoning ability (Liu et al., 2023a; Sun et al., 2022).

In an endeavor to comprehensively understand the capabilities of LLMs, numerous studies have assessed their performance across a variety of language tasks (Zhang et al., 2023; Beltagy et al., 2022). These tasks include reasoning (Bang et al., 2023; Xu et al., 2023), machine translation (Jiao et al., 2023), and question-answering (Tan et al., 2023). Furthermore, the scope of research has been expanded to encompass broader domains. For instance, the applicability of LLMs in AI-assisted medical education has been explored through their ability to answer questions from medical licensing exams Kung et al. (2023); Patel & Lam (2023); Lu et al. (2022). Collectively, these studies suggest that LLMs have the potential to achieve new state-of-the-art performance in traditional tasks and can even establish a new paradigm of research based on interactions with a language model.

So far, a wide range of language models have achieved a great success for bioinformatics tasks (Otmakhova et al., 2022), such as evolutionary scale modeling (ESM) (Lin et al., 2023) and pre-trained models for proteins (Elnaggar et al., 2021). These pre-trained models can be used to predict structure, functionality, and other protein properties, or convert proteins into embedding for downstream tasks. For example, AMP-BERT (Lee et al., 2023) is a fine-tuned model by leveraging protein language model (Elnaggar et al., 2021), which achieves remarkable performance in antimicrobial peptide function prediction. However, these previous

studies often utilize pre-trained language models specific to their domain, and hence they may not be powerful as modern LLMs that are trained using a wide-ranging corpus of text. Moreover, their research typically concentrates on a limited set of tasks, resulting in a lack of systematic and comprehensive investigations into the potential of LLMs for broader bioinformatics research. Evaluating bioinformatics tasks using LLMs can offer a new, effective approach to understanding and solving complex bioinformatics problems, and thus is a research direction of great significance.

In this work, we investigate the potential applications of LLMs on several popular bioinformatics tasks. Our investigation includes a diverse set of tasks that provide a comprehensive evaluation of LLMs within the context of bioinformatics. These tasks comprise the identification of potential coding regions, extraction of named entities for genes and proteins, detection of antimicrobial and anti-cancer peptides, molecular optimization, and addressing educational problems within bioinformatics. To conduct our experiments, we represent chemical compounds, DNA and protein sequences in text format and convert the problem in natural language processing. Then, we feed them into LLMs to generate predictions. Our experiments indicate that, given appropriate prompts, LLMs can partially solve these tasks, underscoring the potential utility of LLMs in bioinformatics research. Further analysis of the extensive evaluation leads to three observations. Firstly, with appropriate prompts, LLMs can achieve performance on par with competitive baselines for simple bioinformatics tasks. Secondly, the model has difficulties when faced with more complex tasks; for instance, it may generate non-existing gene name mentions for gene and protein named entity recognition. Lastly, some prompts and model variants could lead to fluctuating the performance, which indicates their choices would require further investigation. By shedding light on the strengths and limitations of LLMs in bioinformatics, we hope this work can enhance the utility of LLMs in supporting data-driven research and problem solving within bioinformatics and pave the way for future research directions.

## 2 Related Work

### 2.1 Large Language Models (LLMs)

With the development of ChatGPT, LLMs have become popular in the artificial intelligence community, which involve billions of parameters. Among various LLMs, ChatGPT has made impressive impacts by showing remarkable performance in zero-shot human-machine interaction (Jahan et al., 2023b). The GPT-3.5 and GPT-5 are the two mainstream models that ChatGPT currently offers. Natural language and code can be understood and produced by the GPT-3.5 models. There may not be much of a difference between GPT-3.5 and GPT-4 in normal speech. When the task's complexity reaches a certain level, though, GPT-4 distinguishes itself from GPT-3.5 by being more dependable, inventive, and capable of handling far more complex instructions[1,2]. Some new models, such as Llama 2 (70B) (Touvron et al., 2023) and Google bard (AYDIN, 2023), have been proposed, and their performance remains to be tested. To understand the capacity and limitations of LLMs, the evaluation of these LLMs has received extensive interest, especially on NLP tasks such as reasoning (Bang et al., 2023), machine translation (Jiao et al., 2023) and question answering (Tan et al., 2023). For example, ChatGPT has shown extraordinary performance on zero-shot dialogue understanding with the help of proper prompts (Wysocka et al., 2023; Liu et al., 2023b). Although ChatGPT suffers from several limitations in several tasks, the evaluation can also help the enhancement of ChatGPT as the version is updated. However, existing works on the evaluation of LLMs mostly focus on NLP tasks (Biswas, 2023; Shyr et al., 2023; Li et al., 2023; Pu & Demberg, 2023) while the evaluation for Bioinformatics is still underexplored. As a result, we aim to evaluate LLMs on a range of bioinformatics tasks to give insights to the AI for science community.

### 2.2 Language Models for Bioinformatics

Bioinformatics has been an important field involving the collection and analysis of biological data such as DNA sequences and protein structures. Language models have been applied to solve various bioinformatics tasks, such as transforming amino acids into different embeddings using protein language models, which

---

[1]https://openai.com/research/gpt-4

[2]https://platform.openai.com/docs/guides/fine-tuning

can be used for downstream protein understanding. Recently, one of the most impressive applications is AlphaFold2 (Cramer, 2021), which employs a transformer architecture to predict protein structures from amino acid sequences. It has demonstrated remarkable accuracy in predicting protein folding, outperforming traditional methods and highlighting the potential of large-scale models in this domain. Language models have inspired the development of drug discovery and design tasks as well (Deng et al., 2022). For instance, inspired by the success of BERT (Devlin et al., 2018), MolBERT (Fabian et al., 2020) is developed for predicting molecular properties and generating novel molecular structures. MolBERT demonstrates the capacity of language models to generate de novo molecules and predict their physicochemical properties, which can be invaluable in the drug discovery process. In this paper, we study the power of LLMs to directly solve various bioinformatics problems, which can inspire the development of AI for science.

We also notice that there are several works on the evaluation of LLMs related to bioinformatics (Shue et al., 2023; Piccolo et al., 2023). Piccolo et al. (2023) investigate the performance of ChatGPT on bioinformatics programming tasks in educational settings. Shue et al. (2023) produce a tool based on ChatGPT for bioinformatics education. However, these works are a preliminary exploration of ChatGPT for bioinformatics beginners while our work goes deeper and explores more complicated and typical tasks. Jahan et al. (2023a) primarily focuses on the evolution of biomedical text processing problems in large language models. In contrast, our work involves a broad bioinformatics field, including identifying potential coding regions of viruses, detecting antimicrobial and anticancer peptides, gene and protein-named entity extraction, molecular modification, and educational bioinformatics problem-solving sets. Taylor et al. (2022) introduce a large language model for scientific knowledge mining while our work focuses on the evaluation of GPT models on various bioinformatics problems. Luo et al. (2022) design a domain-specific language model for biomedical problems while our work explores more areas in bioinformatics.

We believe our work can help gain a more profound understanding of LLMs' potential in advancing the field of bioinformatics, opening new avenues for data analysis, hypothesis generation, and further automation of complex computational tasks in this area.

## 3 Evaluated Tasks

### 3.1 Identifying Potential Coding Regions from DNA Sequences

The coding sequence (CDS) (Lubahn et al., 1989; Zhu et al., 2010) are some regions of a DNA or RNA sequence that contain essential information for protein encoding. Identifying these potential coding regions within viral sequences could be a crucial task, as it can aid us in understanding the biological characteristics of the corresponding virus as well as the expression of DNA sequences and genes (Badger & Olsen, 1999). We formalize the task as follows:

**Task 1. (Identifying Coding Regions)** *Our objective is to leverage a machine learning approach to identify as many potential coding regions as possible in the given DNA sequence.*

**Prompts:** *Based on your knowledge, describe as many possible potential coding regions if they exist in frame 1.*

### 3.2 Identifying Antimicrobial Peptide

Antimicrobial resistance poses a significant threat to public health (Wise et al., 1998). As a potential solution, antimicrobial peptides (AMPs) (Bahar & Ren, 2013) have emerged as a promising solution, due to their broad-spectrum mechanism of action. Typically, AMPs exterminate bacteria and other hazardous organisms through interfering with vital biological components, such as the cell membrane as well as DNA replication mechanisms (Zasloff, 2002). Therefore, the identification of candidate peptides with antimicrobial functions is crucial for developing novel therapeutics. The task is formalized as follows:

**Task 2. (Identifying Antimicrobial Peptide)** *Given the training set, we train a machin learning model which can identify antimicrobial peptides in massive protein sequences.*

**Prompts:** *You are a peptide design researcher. Please tell me if the given peptide sequence has antimicrobial properties.*

### 3.3 Identifying Anti-cancer Peptide

The majority of anti-cancer drugs have inadequate selectivity, killing both normal and cancer cells without discrimination (Liu et al., 2015). However, anti-cancer peptides (ACPs) function as molecularly targeted peptides that can directly bind to specific cancer cells or organelle membranes, or as binding peptides associated with anti-cancer drugs(Li et al., 2011). As minuscule peptides contain sequences of amino acids, ACPs are cancer-selective and toxic (Tyagi et al., 2015). Therefore, they have emerged as a novel therapeutic strategy that targets some cancer cells specifically (Chiangjong et al., 2020). Considering the extensive time and high costs associated with identifying ACPs through biochemical experimentation, the development of deep learning algorithms for ACPs identification is vital. We formalize the task (Li et al., 2020) as follows:

**Task 3. (Identifying Anti-cancer Peptide)** *Given a training set, we train a machine learning model which can identify anti-cancer peptides in these massive protein sequences.*

**Prompts:** *You are a peptide design researcher. Please tell me whether a peptide with a sequence: N could be an anti-cancer peptide.*

### 3.4 Molecule Optimization

To evaluate the extent of knowledge that GPTs possess in the realms of chemistry and pharmacology, we study their proficiency in optimizing molecular properties. The optimization of molecules represents a pivotal phase in the process of drug discovery (Verdonk & Hartshorn, 2004), allowing for enhancing the desired characteristics (e.g., octanol-water partition coefficients (Sangster, 1997), synthetic accessibility (Ouyang et al., 2021) and drug-likeness (Bickerton et al., 2012)) of drug candidates via targeted chemical modifications. The task is formalized as follows:

**Task 4. (Molecule Optimization)** *Our objective is to modify a given molecule while preserving the primary molecular scaffold, such that certain properties can be enhanced.*

**Prompts:** *Assume that you were a medicinal chemist, please make big modifications that go beyond just changing the charge to the following molecule to optimize the octanol–water partition coefficient penalized by synthetic accessibility and ring size. Here's the SMILE string for the molecule, $SMILES$, and output the optimized SMILE string, please.*

### 3.5 Gene and Protein Named Entities Extraction

For genes and proteins, a wide variety of alternative names are used in abundant scientific literature or public biological databases, which poses a significant challenge to the gene and protein named mentions finding task. Meanwhile, as new biological entities are continuously discovered (Bruford et al., 2020; Fundel & Zimmer, 2006; Rindflesch et al., 1999; Blaschke et al., 2002), the diversity of gene and protein nomenclatures also brings some challenges to the gene and proteins named mention extraction task. For instance, the gene names in Drosophila (Yeh et al., 2005), a.k.a., the fruit fly (Wangler et al., 2015), can be common English words such as *white, dumpy and forked.* This nature could lead to a misleading algorithm or recognition model, making it a challenge to accurately extract gene and protein names. Here, we formalize the task as follows:

**Task 5. (Gene and Protein Named Entities Extraction)** *Our objective is to identify potential gene and protein named entities from the given sentences in life science literature.*

**Prompts:** *You are an expert in the Named Entity Recognition field. Given a token and a sentence that contains gene mentions, you are to generate an ASCII list of identified gene names. Each gene mention will be formatted as follows: sentence-identifier | start-offset end-offset | optional text. Each gene mention from the same sentence will be listed on a separate line. If a sentence doesn't have any gene mentions, it won't be included in the list. Counting numbers need to exclude spaces, sentence-identifiers, and start from 1.*

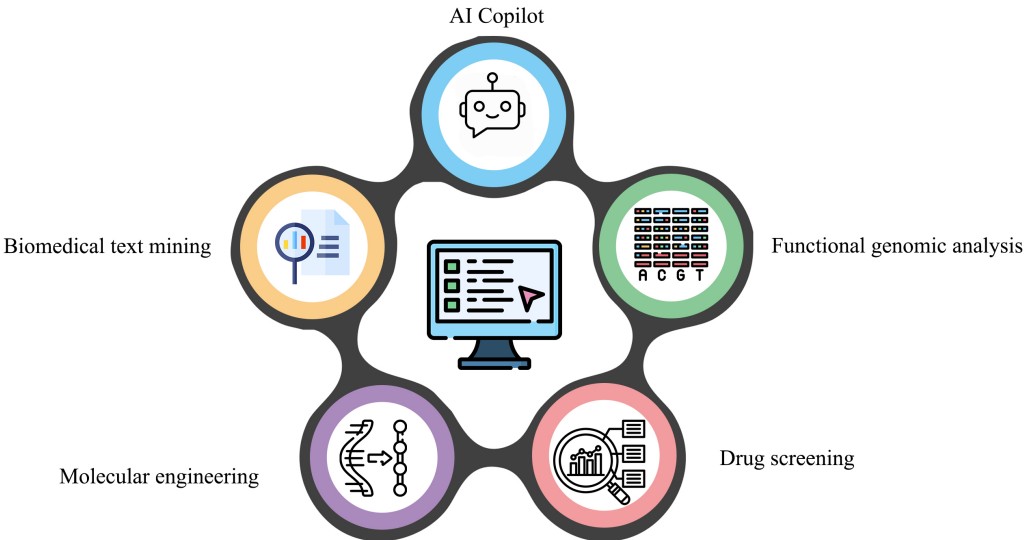

Figure 1: Our tasks aim to validate how LLMs can benefit bioinformatics research.

### 3.6 Evaluation on Educational Bioinformatics Problem-Solving Set

In addition to using various datasets and different case scenarios to demonstrate the performance of GPTs, we also employ a set containing 105 bioinformatics questions for evaluation. These problems originate from Rosalind[3], an educational platform dedicated to learning bioinformatics and programming through problem solving. The task is formalized as follows:

**Task 6. (Educational Bioinformatics Problem Solving)** *Our objective is to generate corresponding answers to a bioinformatics problem set. These problems primarily encompass seven topics, i.e., String Algorithms, Combinatorics, Dynamic Programming, Alignment, Phylogeny, Probability, and Graph Algorithms.*

**Prompts:** *Each question will be provided as a prompt.*

As shown in Figure 1, we analyze the significance of studying these tasks for bioinformatics research: Task 1 focuses on the analysis of genome sequences to identify potential coding regions, indicating that LLMs are promising for functional genomic analysis. Task 2 and Task 3 are about AMP and ACP, which can validate that LLMs can contribute to drug screening. Task 4 aims to introduce LLMs for molecular engineering, indicating that understanding the complex interplay between sequence patterns and biological functions allows LLMs to contribute to the design of optimized proteins and molecules. Furthermore, as showcased in Task 5, LLMs can perform entity recognition via specific prompts, aiding in biomedical text mining. Lastly, as illustrated by Task 6, LLMs can provide assistance to bioinformatics challenges, exemplifying their practical value in the field.

We also show the task details and datasets in Table 6. For Task 1, we utilize GPT-3.5 and GPT-4 to validate the basic biological knowledge of the language model. We additionally introduce Llama 2 and Google Bard for validation. For Tasks 2 and 3, we use GPT-3.5 (Davinci-ft), ESM, and AMP-BERT to test how well they can predict antimicrobial and anticancer peptides. GPT-4 does not support fine-tuning, which is skipped in our experiments. For Task 4, we use GPT-4 to verify the model's capability for molecular modification. A baseline Modof is introduced for molecule optimization. GPT-3.5 is skipped because of its worse performance. For Task 5, we employ GPT-4 and GPT-3.5 (gpt-3.5-turbo-0613) to assess the model's recognition ability for genes and protein names. BioBERT, MT-BERT, and MT-BioBERT are additionally used for named-entity

---

[3]https://rosalind.info/problems/list-view/

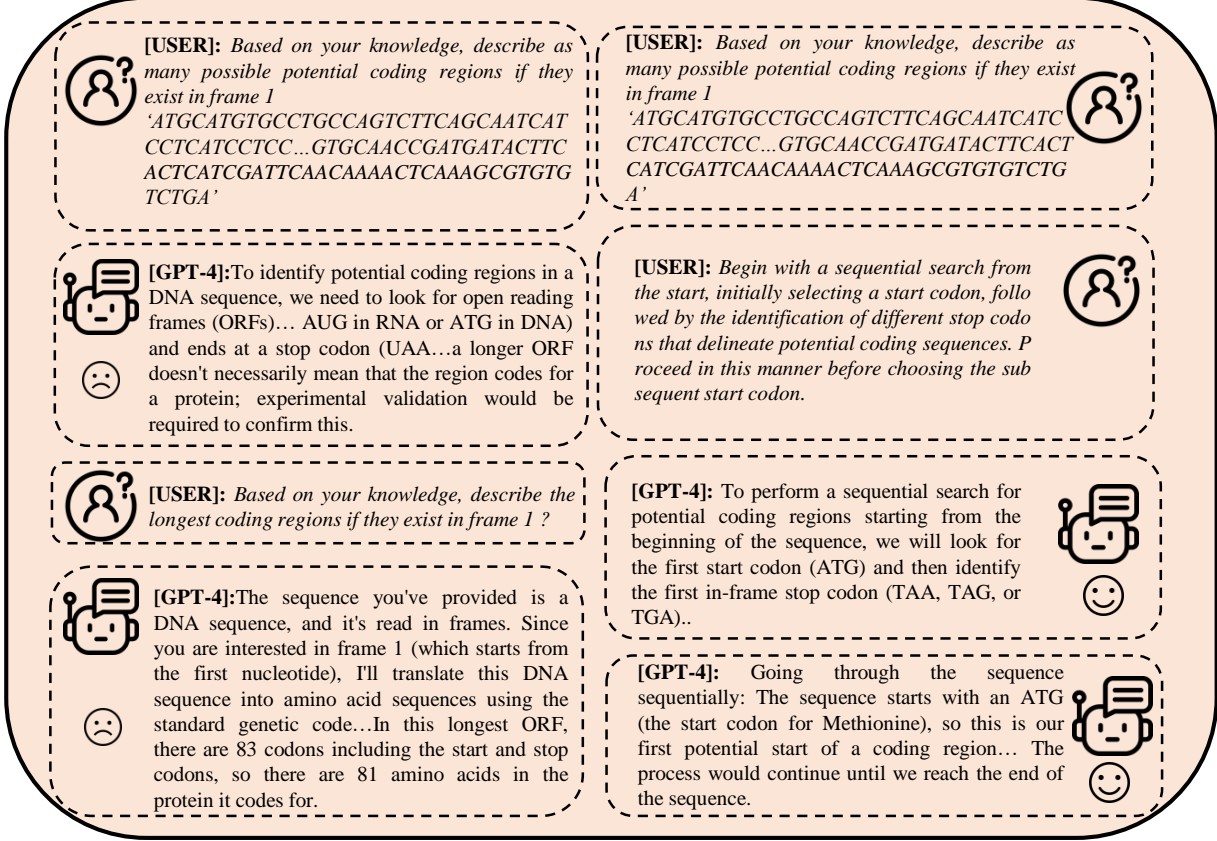

Figure 2: Illustration of the identification of coding regions utilizing GPT-4. (The dialogue on the right depicts a comparison using chain of thought.)

recognition. For Task 6, we verify the model's ability of GPT-3.5 and GPT-4 to answer biological questions involving probabilistics, logic, and character processing.

## 4    Results and Discussions

### 4.1    Performance on Identifying Potential Coding Regions

As for identifying potential coding regions (CDS) (Furuno et al., 2003) from DNA sequences, we utilize the understanding abilities of GPT-4, GPT-3.5, Llama 2 (70B) (Touvron et al., 2023) and Google bard (AYDIN, 2023) with a corresponding prompt. Since LLMs have been trained in a variety of internet texts, the analyzing capability of LLMs allows them to generate meaningful and contextually appropriate responses.

Our test subject is the DNA sequence of the Vaccinia virus (Goebel et al., 1990) using partial sequence (GeneID: 3707616, 3707624, and 3707625. ACCESSION Id: NC_006998.1), and we require LLMs to give potential CDS in the first frame. The results are shown in Figure 2 and Figure 5. From the results, GPT-4 can successfully deliver the definitions of CDS and Open Reading Frames (ORFs), accurately pinpointing the start codon (usually AUG in RNA or ATG in DNA) and stop codons (UAA, UAG, UGA for RNA or TAA, TAG, TGA for DNA). It generates a list of potential coding regions, specifying their nucleotide lengths and corresponding start and stop codons.

- *The performance of GPT-4 can be enhanced by adopting a thought-chain approach.* Despite its superior advantage, GPT-4 still overlooks some potential coding regions. When tasked with identifying the longest coding region in the first frame, it begins by translating the DNA sequence into an amino acid sequence

Table 1: The compared cross-validation results of different models for identifying antimicrobial peptides on the training set.

| MODEL | SN | SP | F1 | ACC | AUROC | AUPR |
|---|---|---|---|---|---|---|
| XGB | 0.702 | 0.566 | 0.630 | 0.641 | 0.734 | 0.783 |
| MNB | 0.815 | 0.739 | 0.780 | 0.800 | 0.870 | 0.912 |
| SVM | 0.872 | 0.717 | 0.790 | 0.796 | 0.843 | 0.897 |
| KNN | 0.709 | 0.622 | 0.670 | 0.703 | 0.674 | 0.722 |
| LR | 0.843 | 0.735 | 0.790 | 0.804 | 0.836 | 0.889 |
| MLP | 0.792 | 0.654 | 0.720 | 0.731 | 0.776 | 0.802 |
| RF | 0.867 | 0.691 | 0.770 | 0.772 | 0.834 | 0.870 |
| GB | 0.775 | 0.583 | 0.660 | 0.646 | 0.708 | 0.789 |
| ESM | 0.912 | 0.928 | 0.920 | 0.920 | **0.974** | 0.977 |
| AMP-BERT | 0.926 | 0.930 | 0.928 | 0.928 | 0.966 | 0.965 |
| GPT-3.5(Davinci-ft) | **0.979** | **0.962** | **0.970** | **0.968** | 0.968 | **0.978** |

Table 2: The compared results of different models for identifying antimicrobial peptides on the test set.

| MODEL | SN | SP | ACC | Fl | AUC | AUPR |
|---|---|---|---|---|---|---|
| XGB | 0.695 | 0.630 | 0.660 | 0.654 | 0.714 | 0.700 |
| MNB | 0.687 | **0.750** | 0.711 | 0.746 | 0.757 | 0.720 |
| SVM | 0.740 | 0.675 | 0.706 | 0.702 | 0.749 | 0.697 |
| KNN | 0.608 | 0.687 | 0.632 | 0.698 | 0.691 | 0.738 |
| LR | 0.724 | 0.676 | 0.699 | 0.702 | 0.746 | 0.711 |
| MLP | 0.701 | 0.715 | 0.707 | 0.730 | 0.749 | 0.715 |
| RF | 0.714 | 0.692 | 0.703 | 0.715 | 0.739 | 0.692 |
| GB | 0.708 | 0.606 | 0.646 | 0.616 | 0.699 | 0.691 |
| ESM | 0.865 | 0.496 | 0.742 | 0.688 | 0.779 | 0.758 |
| AMP-BERT | **0.876** | 0.635 | **0.792** | **0.760** | **0.818** | 0.787 |
| GPT-3.5 (Davinci-ft) | 0.844 | 0.745 | 0.718 | 0.759 | 0.782 | **0.810** |

using the standard genetic code, followed by a detailed explanation of the code. However, it incorrectly identifies an open reading frame (ORF) with 83 codons as the longest CDS. Therefore, we attempt to provide step-by-step prompts to GPT-4: "Begin with a sequential search from the start, initially selecting a start codon, followed by the identification of different stop codons that delineate potential coding sequences. Proceed in this manner before choosing the subsequent start codon." With this approach, GPT-4 is able to present all potential CDS, including the longest sequence.

- *GPT-4 is capable not only of directly providing potential coding sequences but also of delivering an effective algorithm to algorithmically find all potential coding sequences.*, We observe that the GPT-4 program can report all potential coding regions successfully in the given examples.

- *The performance of Llama 2 (70B) is the least satisfactory, failing to yield any potential coding sequences or explanations. In contrast, Google Bard only offers suggestions for identifying potential coding sequences using other tools and points out the start and stop codons in the DNA sequence.* When we request a program from Google Bard to find coding sequences, it is able to provide an effective algorithm.

## 4.2 Performance on Identifying Antimicrobial Peptide

As for identifying antimicrobial peptides, in this subsection, we fine-tune the GPT-3.5 (Davinci) model[4] to distinguish between AMPs and non-AMPs. We set the number of epochs, batch size and learning rate

---
[4]https://platform.openai.com/docs/models/gpt-3

multiplier during training to 20, 3 and 0.3, respectively. Our selected prompt is as follows: *Assuming the role of a peptide design researcher, please evaluate if a peptide with this particular sequence could qualify as an antimicrobial peptide.*

To access the performance of our fine-tuned model GPT-3.5 (Davinci-ft), we conduct a comparison with the advanced protein large language model, ESM (esm_msa1b_t12_100M_UR50S), several machine learning-based methods, i.e., XGBoost (XGB) (Chen et al., 2015), Multinomial Naive Bayes (MNB) (Kibriya et al., 2005), Support Vector Machines (SVM) (Jakkula, 2006), K-Nearest Neighbor (KNN) (Guo et al., 2003), Logistic Regression (LR) (Maalouf, 2011), MultiLayer Perceptron (MLP) (Pinkus, 1999), Random Forest (RF) (Biau & Scornet, 2016), GBoost (GB) (Saigo et al., 2009), and AMP-BERT (Lee et al., 2023) using two datasets from (Lee et al., 2023). We utilize six widely accepted metrics to assess the performance, namely, sensitivity (SN), specificity (SP), F1-score (F1), accuracy (ACC), area under the Receiver Operating Characteristic curve (AUROC), and area under the Precision-Recall curve (AUPR).

The training set (Lee et al., 2023) is comprised of 1,778 AMPs paired with an equal number of non-AMPs. The test set constitutes 2,065 AMPs and 1,908 non-AMPs. Importantly, these two data sets have low overlap. In alignment with the comparison methodology outlined by (Lee et al., 2023), we initially perform a 5-repeated 10-fold cross-validation during the fine-tuning stage on the training set. Subsequently, the GPT-3.5 (Davinci-ft) model is tested to the test set. The results for different models on the training and test sets are shown in Table 1 and Table 2, respectively. From the results, we have the following observations:

- *GPT-3.5 (Davinci-ft) demonstrates the best performance on most metrics during the 5-repeated 10-fold cross-validation process.* Different from AMP-BERT which is based on ProtTrans (Elnaggar et al., 2021) specifically trained on proteins, GPT-3.5 (Davinci-ft) is not a model targeting proteins. Nevertheless, after the fine-tuning procedure, GPT-3.5 (Davinci-ft) outperforms the AMP-BERT model across a variety of metrics. In terms of F1-score, GPT-3.5 (Davinci-ft) also significantly surpasses the other models. Specifically, it outperforms XGB, MNB, SVM, KNN, LR, MLP, RF, GB, and AMP-BERT by margins of 0.340, 0.190, 0.180, 0.300, 0.180, 0.250, 0.200, 0.310, and 0.042, respectively, which validates the strong capacity of LLMs.

- *GPT-3.5 (Davinci-ft) has the potential to tackle the imbalanced test set.* As shown in Table 2, for the metrics of SN, ACC, F1, and AUC, AMP-BERT achieves the best performance with scores of 0.876, 0.792, 0.760, and 0.818, respectively. On the SP metric, MNB demonstrates the best performance with a score of 0.750.

- ESM demonstrates strong performance on the antimicrobial peptide training sets. Specifically, it achieved the best performance on the training set with an AUROC of 0.974, surpassing other models. Though, GPT-3.5 (Davinci-ft) achieves the best performance on AUPR with a score of 0.810. This suggests that it still has the potential in handling the imbalance between positive and negative instances in the test set.

### 4.3 Performance on Identifying Anti-cancer Peptide

As for identifying anti-cancer peptides (ACPs), we also fine-tune the Davinci model, named as Davinci-ft to distinguish between ACPs and non-ACPs. We use the same setting during the training process. Our selected prompt is as follows: *Assuming the role of a peptide design researcher, please evaluate if a peptide with this particular sequence could qualify as an anti-cancer peptide.*

Our dataset originates from (Li et al., 2020) encompassing a total of 138 ACPs and 206 non-ACPs. To evaluate the performance of the GPT-3.5 (Davinci-ft) model, we compare it with several machine learning-based methods utilizing widely accepted metrics as detailed in Section 4.2. The compared results are shown in Table 3. We observe that for the SN metric, MNB, SVM, LR, MLP, RF, GB and GPT-3.5 (Davinci-ft) all achieved top performances. In terms of SP, KNN achieves the highest performance, indicating its superior ability to identify non-ACPs. More importantly, GPT-3.5 (Davinci-ft) achieves the best performance in terms of most matrices including MCC, AUC and AUPRC. In particular, GPT-3.5 (Davinci-ft) displays exceptional performance on the crucial AUC metric, outperforming XGB, MNB, SVM, KNN, LR, MLP, RF,

Table 3: The compared results of different methods for identifying anti-cancer peptides.

| MODEL | SN | SP | ACC | MCC | AUC | AUPRC |
|---|---|---|---|---|---|---|
| XGB | 0.846 | 0.864 | 0.857 | 0.700 | 0.806 | 0.845 |
| MNB | **1.000** | 0.913 | **0.943** | 0.885 | 0.973 | 0.973 |
| SVM | **1.000** | 0.875 | 0.914 | 0.829 | 0.966 | 0.964 |
| KNN | 0.929 | **0.952** | **0.943** | 0.881 | 0.930 | 0.943 |
| LR | **1.000** | 0.840 | 0.886 | 0.775 | 0.963 | 0.962 |
| MLP | **1.000** | 0.913 | **0.943** | 0.885 | 0.949 | 0.958 |
| RF | **1.000** | 0.875 | 0.914 | 0.829 | 0.990 | 0.984 |
| GB | **1.000** | 0.778 | 0.829 | 0.667 | 0.857 | 0.875 |
| ESM | 0.933 | 0.900 | 0.903 | **0.914** | 0.923 | 0.920 |
| GPT-3.5 (Davinci-ft) | **1.000** | 0.875 | 0.914 | 0.892 | **0.993** | **0.991** |

Table 4: The average property improvements of the whole molecules dataset by GPT-4 and Modof.

| Method | $\Delta$logP | $\Delta$SA | $\Delta$QED |
|---|---|---|---|
| Modof | **3.76** | 0.20 | -0.19 |
| GPT-4 | 1.87 | **0.84** | **0.03** |

and GB by respective margins of 0.187, 0.020, 0.027, 0.063, 0.030, 0.044, 0.003, and 0.136. These results collectively indicate that LLMs can achieve superior performance in identifying anti-cancer peptides.

### 4.4 Performance on Molecule Optimization

As for molecule optimization, we target at the enhancement of partition coefficients, a.k.a., logP, which can be quantified by Crippen's logP methodology (Wildman & Crippen, 1999). Meanwhile, we also consider penalties incurred by synthetic accessibility (SA) (Ertl & Schuffenhauer, 2009). Following Bickerton et al. (2012), we also evaluate quantitative results in terms of drug-likeness (QED) scores, which can reflect whether a molecular can be a drug candidate or not. To evaluate the performance of GPT-4, we present a comparative analysis with Modof (Chen et al., 2021), a sophisticated deep generative model. Modof harnesses the capabilities of the junction tree methodology for molecular representation, modifying discrete fragments of the molecule through the employment of variational autoencoders (VAE) (Jin et al., 2018). Here, the dataset, we used, is originates from the ZINC database (Sterling & Irwin, 2015).

We summarize the quantitative results in Table 4 and Figure 4, and some examples can be found in Table 7. We observed a significant difference, as several molecules modified by GPT-4 and Modof exhibit on the metrics. The p-values (T-test) for $\Delta logP$, $\Delta QED$, and $\Delta SA$ are $4.86 \times 10^{-65}$, $1.89 \times 10^{-72}$, and $1.34 \times 10^{-38}$, respectively. The mean value for Modof $\Delta \log P$ is 3.76, with a 95% confidence interval ranging from 3.58 to 3.93. For GPT-4, the mean of $\Delta SA$ is 0.198, with a confidence interval from 0.136 to 0.257. Lastly, the mean for $\Delta QED$ is -0.194, with its confidence interval lying between -0.213 and -0.175. From the results, we have the following observations:

- *GPT-4 can successfully generate valid SMILES in most cases.* It becomes evident that it has assimilated fundamental principles of physical chemistry. GPT-4 can provide valid optimized molecules for 661 cases, which achieves the comparable performance compared with the superior baseline Modof. In particular, the validity rateof Modof and GPT-4 are 0.800 and 0.830, respectively.

- *GPT-4 has an advantage in improving both SA and QED.* GPT-4 can achieve higher scores in terms of both SA and QED in most cases. Remarkably, owing to its advanced conversation diagram, GPT-4 is capable of articulating the executed modifications, providing rudimentary rationales behind the alterations.

- *GPT-4 still falls short in improving logP.* The average $\Delta$ logP achieved by GPT-4 is about 50.3% of that achieved by Modof. We have computed the deviation of the heavy atom number before and after

Table 5: The compared results of different models for extracting gene and protein name mention in GM test set.

| Model | P | R | F |
|---|---|---|---|
| BiLSTM | 87.98 | 88.25 | 88.11 |
| MT-BiLSTM-CRF | 82.10 | 79.42 | 80.74 |
| BioBERT | 84.32 | 85.12 | 84.72 |
| MT-BERT | 84.12 | 84.98 | 84.53 |
| MT-BioBERT | 84.53 | 85.27 | 84.82 |
| GPT-3.5(gpt-3.5-turbo-0613) | 24.07 | 7.26 | 11.17 |
| GPT-4 | 51.72 | 77.7 | 62.10 |

molecular optimization. The average deviation of heavy atom number for Modof optimized molecules is +10.26, while the average deviation of heavy atom number for GPT-4 optimized molecules is only +0.65, which is much lower compared with Modof optimization. The potential reason is that GPT-4 would tend to remove atomic charges in an attempt to improve the octanol-water partition coefficient instead of modifying the hydrophobic fragments into hydrophilic ones. Such a practice could potentially obliterate significant pharmacophores of a drug. Furthermore, GPT-4 often adopts a more conservative approach to modifying the original molecule, primarily excising some fragments to facilitate synthesis, whereas Modof typically opts to append larger new fragments to the molecule. As a consequence, the simple and direct modification learned by GPT-4 could be insufficient for improving logP significantly as Modof does.

### 4.5 Performance on Gene and Protein Named Entity Recognition

As for gene and protein named entity recognition (NER), we evaluate the performance of GPT-3.5 on BioCreative II gene mention (GM) corpus (Yeh et al., 2005). This dataset comprises extensive annotated sentences from MEDLINE (Greenhalgh, 1997), with a primary goal to focus on the extraction of gene and protein named entities. Here, the test set contains 5,000 sequences. We leverage a prompt to call a GPT-3.5 API (gpt-3.5-turbo-0613) and GPT-4, which evaluates the performance on the test set without utilizing the training set. We compare the API model with the baseline BiLSTM (Cho & Lee, 2019),MT-BiLSTM-CRF (Zhao et al., 2018), BioBERT (Lee et al., 2020), MT-BERT (Cho & Lee, 2019), and MT-BioBERT (Bansal et al., 2020) using three metrics. The compared results are shown in Table 5, and we observe that the API model would achieve poor performance for extracting genes or proteins from sequences in terms of both partial and strict matching criteria. The former requires the prediction should exactly match the ground truth while the latter allows partially overlaps (Zhang et al., 2021). Here, we analyze two limitations of the API model:

- *The GPT-3.5 model could miss gene mention entities in sentences.* As in Figure 6, in the cases beginning by BC2GM001536665 and BC2GM002436660, the ground truth is *"ERCC3Dm protein; 'helicase' domains"* and *"Htf9-a gene; RanBP1 protein; Ran GTPase"*, respectively. The GPT-3.5 model misses *"'helicase' domains"* and reports *"Htf9-a gene; RanBP1; Ran GTPase"*, which results from the confounding gene names in the large corpus.

- *The GPT-3.5 could misunderstand the gene name.* In the case beginning by BC2GM062242948, the ground truth is supposed to be *"AD1 Ag"* while the GPT-3.5 model reports *"AD1; Ag; 2H3"*, which validates that the API model cannot understand that *"AD1 Ag"* should be considered as a whole rather than separately. GPT-3.5 has very low scores, with an F1-Score of 11.17%.

- *GPT-4 achieves better performance than GPT-3.5.* GPT-4 has better performance than GPT-3.5 but still lower than the other models, with an F1-Score of 62.10%.

- *The BiLSTM model outperforms the others with the highest F1-Score, while MT-BiLSTM-CRF scores lower, and BioBERT variants show comparable results.* The BiLSTM model has the highest scores in all three categories, with the F1-Score at 88.11%. The MT-BiLSTM-CRF model has lower scores than

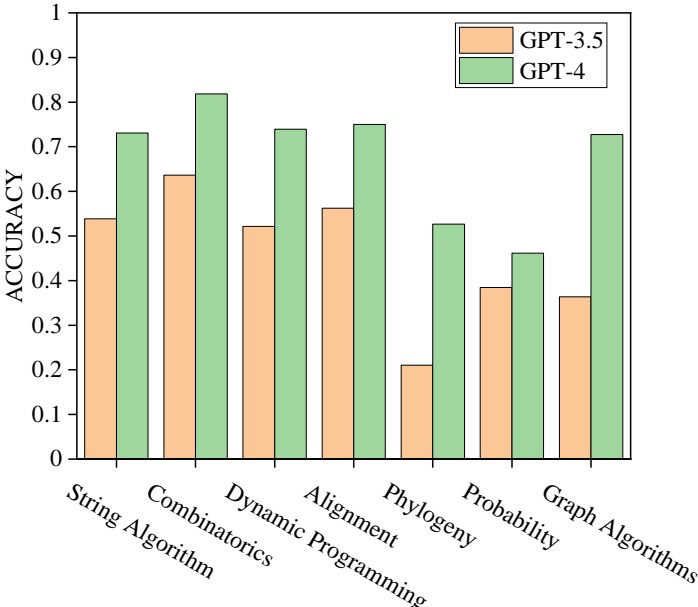

Figure 3: Comparative performance of GPT-3.5 and GPT-4 in various bioinformatics problem-solving tasks.

BiLSTM, with an F1-Score of 80.74%. BioBERT, MT-BERT, and MT-BioBERT models have similar results, with F1-Scores around 84-85%.

### 4.6 Performance on Educational Bioinformatics Problem Solving

As for the evaluation of educational bioinformatics problem solving, we check the performance of two GPT models, GPT-3.5 and GPT-4, across a set of 105 varied problems within the Bioinformatics Stronghold. This collection primarily encompasses seven basic topics as follows: (1) String Algorithms (Baeza-Yates, 1989): This topic focuses on the manipulation and exploration of properties inherent to symbol chains. (2) Combinatorics (Lovász & Prömel, 2004): This scope of problems quantifies distinct objects mathematically. (3) Dynamic Programming (Bellman & Dreyfus, 2015): This topic involves progressively building up solutions to complex problems. (4) Alignment (Edgar & Batzoglou, 2006): This process superimposes symbols of one string over another, inserting gap symbols into the strings as necessary to represent insertions, deletions, and substitutions. (5) Graph Algorithms (Even, 2011): This field involves interpreting and manipulating network structures or graphs. (6) Phylogeny (Field et al., 1988): This topic models the evolutionary trajectories of taxa. (7) Probability (Grinstead & Snell, 1997): This branch of mathematics studies the likelihood of random event occurrences.

We manually collect these 105 problems, individually take them as chat dialogue, and put them into GPT-3.5 and GPT-4. We use the example data set for querying and assessing accuracy. The results related to the seven distinct topics are presented in Figure 3. For a more intuitive illustration, we also select several examples to demonstrate the incorrect and correct responses from GPT-3.5 and GPT-4, as well as their differing performances on the same problems in Figure 7. From the results, we can make the following four observations:

- *GPT-4 demonstrates an overall improvement compared to GPT-3.5 across various types of problems.* This is especially evident in *combinatorics and graph algorithms*, where GPT-4 achieves success rates of 81.82% and 72.73% respectively, substantially higher than those achieved by GPT-3.5. Even in Phylogeny and Probability, where both models showed relatively lower performance, GPT-4 again leads with 52.63% and 46.15% success rates, outperforming GPT-3.5's 21.05% and 38.46%. These findings, collated from a

total of 105 problems, highlight the superior capabilities of GPT-4 in a broad spectrum of bioinformatics challenges, answering 71 questions correctly, compared to GPT-3.5 which correctly solves 53 questions.

- *The performance of GPT-3.5 and GPT-4 is almost consistent across different topics.* Simultaneously, we notice that for probability-related problems, both GPT-3.5 and GPT-4 could only achieve accuracy rates of 36.4% and 46.2% respectively. In contrast, both models are good at solving Combinatorics-related problems, achieving higher accuracy rates of 63.3% for GPT-3.5 and 81.8% for GPT-4.

- *GPT-3.5 exhibits excellent performance in handling relatively simple problems.* As illustrated in Figure 7, when tasked with finding common segments between two DNA sequences, GPT-3.5 is able to respond accurately, providing appropriate program solutions and methods. However, when faced with more complex problems, such as determining the maximum local alignment score of two protein strings, GPT-3.5 could propose incorrect results which are seemingly correct.

- *GPT-4 demonstrates a limited capacity in tackling complex problems.* For instance, as illustrated in Figure 7, when given the exons and introns of a DNA string, we test GPT-4 by deleting the introns, concatenating the exons to form a new string, and then transcribing and translating this newly formed string. GPT-4 successfully manages to remove the introns and translate the DNA into an amino acid sequence. However, for complex logical problems, such as calculating the number of all basepair edges in a bonding graph that can be exactly matched, GPT-4 manages to outline a correct approach but ultimately provides an incorrect answer. Perhaps we need multi-turn interactions to thoroughly solve these complicated problems.

## 5 Limitations

A limitation of our GPT evaluation is that we cannot get access to the training data for GPT, and thus cannot guarantee these datasets are not included in pretraining. We will utilize more up-to-date test data in future works to promise that the training material does not include this. Another limitation is that some of the models could be depreciated in the future. However, we believe that the performance would be enhanced with an updated model in the future, and our method focuses on bridging GPT and bioinformatics rather than a specific LLM, which is meaningful to provide guidance for this field. We will also update the results with more advanced LLMs in our future work.

Our work is committed to the executive order regarding the use of LLMs on biological sequences, and we will always ensure the safety of AI work. LLMs have achieved significant process fields, and it is important to explore whether bioinformatics can benefit from this as well, which can provide guidance for bioinformatics researchers. Moreover, we can also provide insights for researchers using AI in science and encourage more AI researchers to contribute to natural science. Although this work makes the evaluation on six basic bioinformatics tasks, a wide range of sub-regions in bioinformatics have not been considered. In the future, we will further test and develop the relevant applications of the GPT model through more enriched text scenarios. This will specifically manifest in the generative functionalization of sequences of large biomolecules, predicting the interactions between large biomolecules or between drugs and their corresponding receptors, designing and functionalizing large biomolecules from scratch based on original wet-lab data, and establishing a bioinformatics application ecosystem through the GPT model.

## 6 Conclusion

This paper explores the applications of GPTs in bioinformatics research and evaluates them in six basic tasks, including gene and protein named entities extraction, solving educational bioinformatics problems and identifying potential coding regions, antimicrobial and anti-cancer peptides. Extensive experiments demonstrate that LLMs like GPTs can achieve remarkable performance on the majority of these tasks with proper prompts and models. We hope this work can facilitate researchers in bioinformatics about using advanced LLMs and thus promote the development of AI for science.

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

# A  Task Summary

Table 6: Summary of bioinformatics tasks and corresponding datasets.

| Task | Task Description | Dataset Source | Dataset Description | Evaluation Model |
|------|-----------------|----------------|---------------------|------------------|
| Task 1 | Identifying Coding Regions | (Goebel et al., 1990) | Coding regions of the Vaccinia virus | GPT-3.5, GPT-4, Llama 2 (70B), Google Bard |
| Task 2 | Identifying Antimicrobial Peptide | (Lee et al., 2023) | The training set is composed of 1,778 antimicrobial peptides (AMPs) and 1,778 non-AMPs, each with an average length of 34 amino acids. The test set contains 2,065 AMPs and 1,908 non-AMPs (each with an average amino acid length of 39), which have a sequence similarity of less than 90% compared to the training set as determined by CD-HIT. | GPT-3.5 (Davinci-ft), ESM, AMP-BERT |
| Task 3 | Identifying Anti-cancer Peptide | (Li et al., 2020) | The dataset comprises 206 non-anticancer peptides and 138 anticancer peptides, and each with an average amino acid length of 25. Peptides exhibiting more than 90% similarity were removed from the dataset using CD-HIT. | GPT-3.5 (Davinci-ft), ESM |
| Task 4 | Molecule Optimization | (Chen et al., 2021) | The test set derived from the Modof dataset contains 800 molecules, with an average molecular weight of approximately 294.27 g/mol as determined through computational analysis of their SMILES representations. | GPT-4, Modof |
| Task 5 | Gene and Protein Named Entities Extraction | (Yeh et al., 2005) (Greenhalgh, 1997) | The whole dataset contains 20,000 sentences (a total of 24,583 gene and protein entities).The test set contains 5,000 sequences. | GPT-4, GPT-3.5 (gpt-3.5-turbo-0613), BioBERT, MT-BERT, MT-BioBERT |
| Task 6 | Educational Bioinformatics Problem Solving | https://rosalind.info/problems/list-view/ | 105 questions were collected from ROSALIND bioinformatics website. | GPT-4, GPT-3.5 |

## B    More Results on Molecule Optimization

Table 7: The result of GPT-4 and Modof in the modification of some molecules.

| Molecule | GPT-4 | | | | Modof | | | |
|---|---|---|---|---|---|---|---|---|
| | Optimized Molecule | $\Delta logP(\uparrow)$ | $\Delta SA(\uparrow)$ | $\Delta QED(\uparrow)$ | Optimized Molecule | $\Delta logP$ | $\Delta SA$ | $\Delta QED$ |
| 1 | | 1.19 | 1.72 | 0.0 | | 1.90 | 0.58 | -0.38 |
| 2 | | 3.1 | 1.24 | 0.15 | | 3.65 | 0.79 | 0.07 |
| 3 | | 1.68 | 0.28 | 0.05 | | 7.81 | -0.54 | -0.57 |
| 4 | | 1.61 | 1.20 | 0.06 | – | - | - | - |
| 5 | – | - | - | - | | 3.79 | 0.35 | -0.26 |
| 6 | | 2.49 | 1.34 | 0.07 | – | - | - | - |
| 7 | | 1.56 | 1.14 | 0.05 | | 1.58 | 2.30 | -0.18 |
| 8 | | 2.04 | 1.63 | 0.05 | | 2.74 | 0.43 | 0.07 |
| 9 | | 0.12 | 1.47 | 0.0 | | -0.04 | 1.29 | -0.11 |
| 10 | – | - | - | - | | 4.70 | 1.38 | -0.14 |
| 11 | | 0.54 | 0.17 | 0.25 | | 2.74 | 0.08 | -0.03 |
| 12 | | -0.46 | 0.17 | 0.28 | | 3.98 | 0.02 | -0.04 |

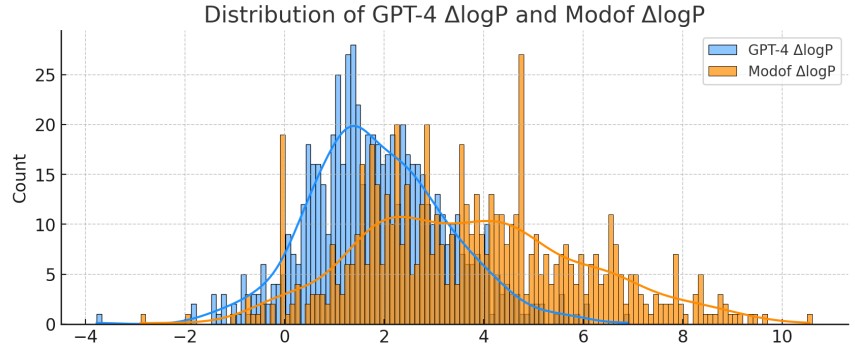

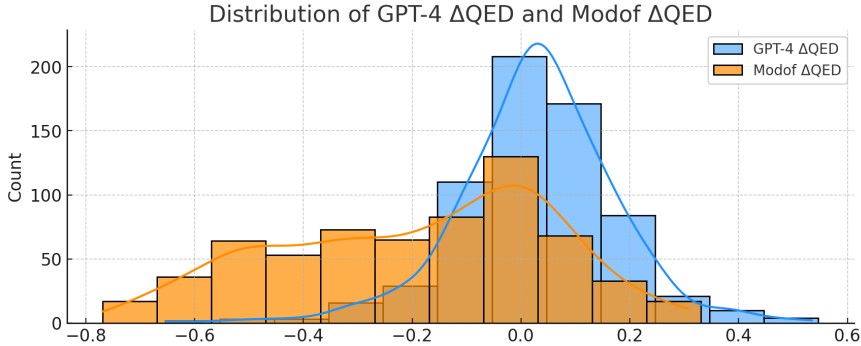

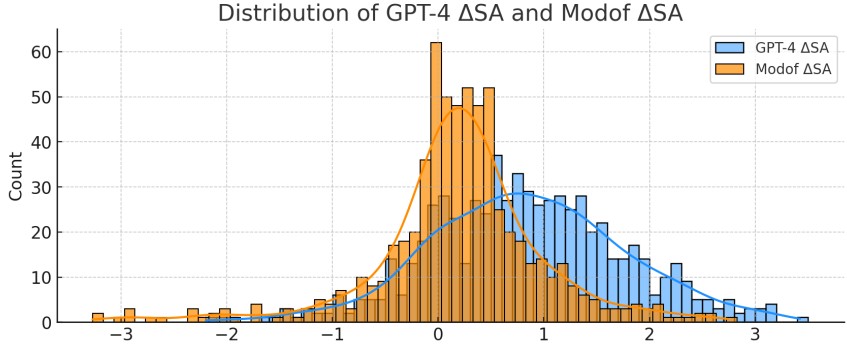

Figure 4: Comparison of distributions for different metrics.

## C  Details of Different Metrics

LogP is a measure of a compound's lipophilicity, indicating its distribution between a hydrophilic (water) and a lipophilic (fat) phase. This property is critical in predicting the absorption, distribution, metabolism, and excretion (ADME) of potential drug candidates. A suitable logP value suggests a balance between solubility (necessary for bioavailability) and permeability (for cellular access). SA assesses the ease with which a compound can be synthesized. This is crucial for practical drug development, as compounds that are difficult or expensive to synthesize may not be viable for large-scale production, regardless of their therapeutic potential. QED measures how closely a compound resembles known drugs based on several physicochemical properties like molecular weight, hydrogen bond donors and acceptors, and logP. This metric helps in prioritizing compounds that have higher chances of success in clinical trials based on historical data. These three metrics are all important for measuring drug properties.

## D    More Results on Identifying Potential Coding Regions from DNA Sequences

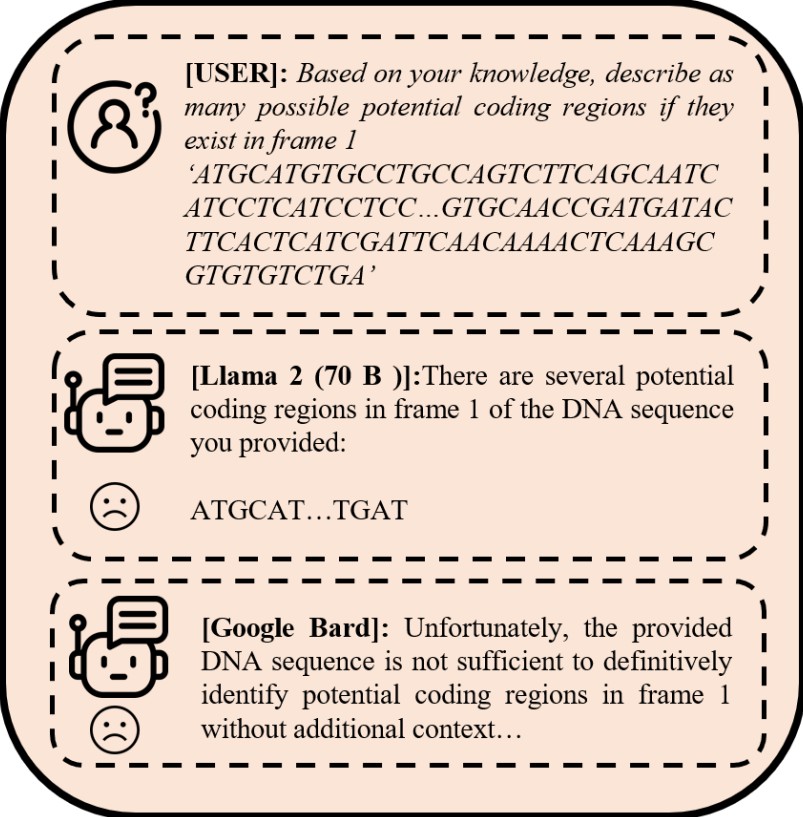

Figure 5:  Comparison of LLMs for identifying potential coding regions from DNA sequences.

## E  More Results on Gene and Protein NER

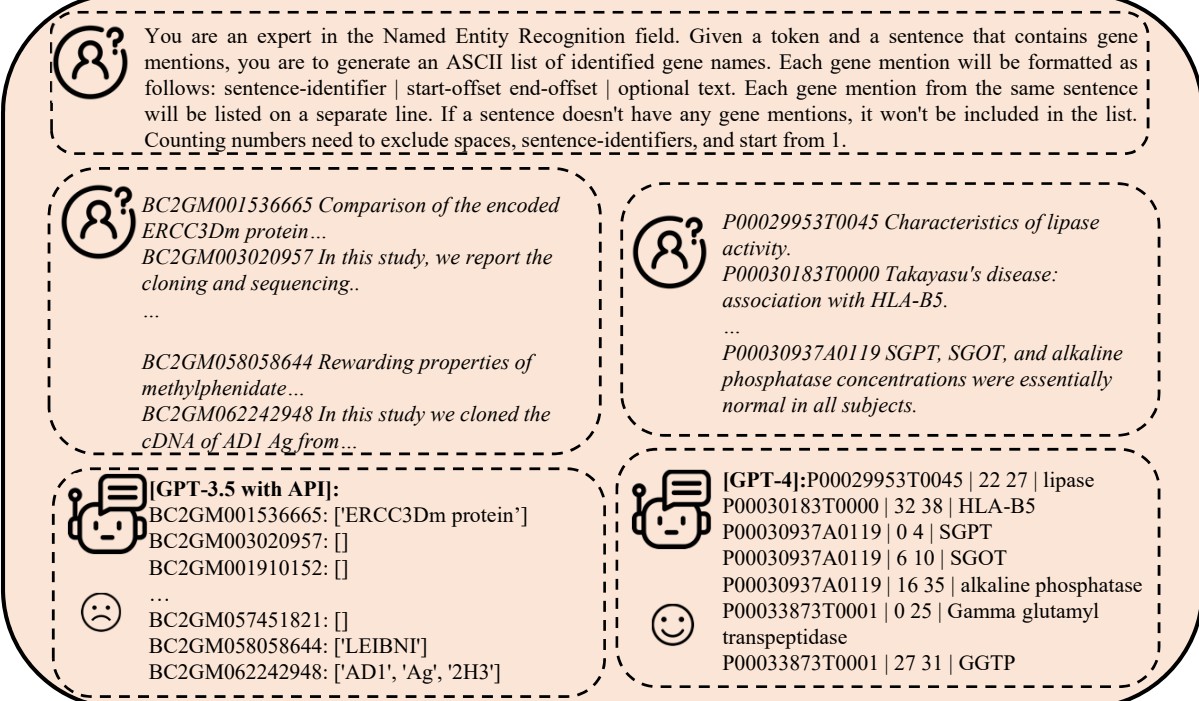

Figure 6: Illustration of gene and protein NER using GPT-3.5 (gpt-3.5-turbo-0613) and GPT-4.

# F   More Results on Bioinformatics Problem Solving

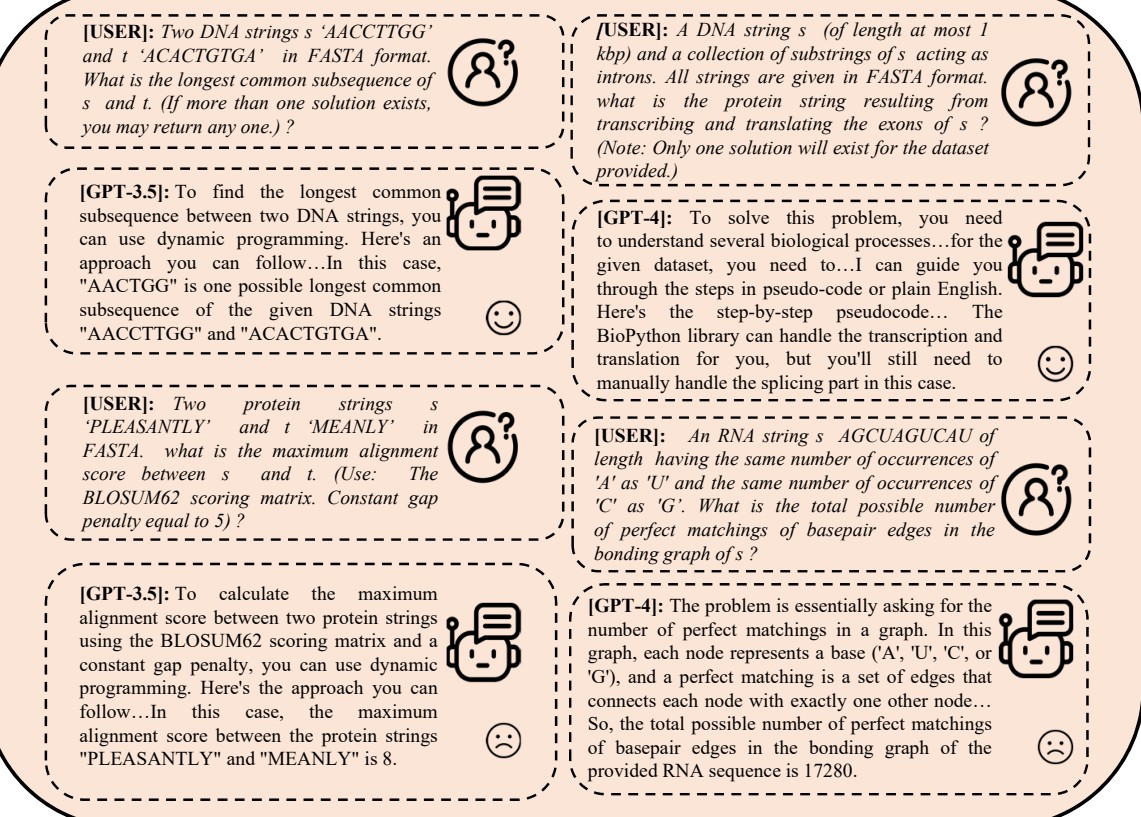

Figure 7: Comparison of GPT-3.5 and GPT-4 in dealing with different tasks.

