# OpenReview forum: "An Evaluation of Large Language Models in Bioinformatics Research"
_TMLR — Rejected by TMLR_

### Review · Reviewer_tQY4 · 2023-11-05

**Summary Of Contributions:**

This work explores the application of large language models (LLMs) such as GPT variants in bioinformatics. It investigates their performance across various bioinformatics tasks, including identifying coding regions, extracting named entities for genes and proteins, detecting antimicrobial and anti-cancer peptides, molecular optimization, and solving educational bioinformatics problems. The findings suggest that with appropriate prompts, LLMs like GPT variants can effectively address many of these tasks. However, the study also highlights the limitations of these models when dealing with complex bioinformatics challenges.

**Audience:**

Yes

**Broader Impact Concerns:**

This work could educate new researchers from the Bioinformatics domain to use Chat-GPT.

**Claims And Evidence:**

Yes

**Requested Changes:**

see weakness

**Strengths And Weaknesses:**

Strengths:
1. The paper is easy to follow.
2. The evaluation is comprehensive.
3. The topic is hot.

Weaknesses:
1. The paper appears to be more of a technical report rather than a traditional research paper. It lacks a substantial discussion related to machine learning, and there are no novel machine learning techniques proposed.

2. There is a similar paper (reference [1]) that is not discussed in this work. The lack of differentiation between the two papers suggests limited novelty in this work.

[1] Jahan, Israt, et al. "A Comprehensive Evaluation of Large Language Models on Benchmark Biomedical Text Processing Tasks." arXiv preprint arXiv:2310.04270 (2023).

3. The role of Chat-GPT in the context of bioinformatics research is unclear. It would be helpful to explain why evaluating Chat-GPT is significant for the bioinformatics research community or specific research groups.

4. It is advisable to include a figure illustrating the relationships among the six tasks mentioned in the paper. Providing context and explaining the relevance of these tasks to bioinformatics research is essential. Do these tasks have specific meaning in bioinformatics?
 Additionally, the paper could explore the possibility of designing new tasks to evaluate Chat-GPT's performance.

---

### Review · Reviewer_FQYe · 2023-11-28

**Summary Of Contributions:**

The authors pose a plethora of bioinformatics problems to a number of OpenAI models, comparing them to standard ML algorithms, as well as OpenAI models of different size.

**Audience:**

Yes

**Broader Impact Concerns:**

Broader Impact Concerns

The White House in the United States recently proposed an executive order regarding the use of LLMs on biological sequences: https://www.whitehouse.gov/briefing-room/presidential-actions/2023/10/30/executive-order-on-the-safe-secure-and-trustworthy-development-and-use-of-artificial-intelligence/

Does this work, or future work, need to be covered by this EA? Why or why not should this be necessary?

**Claims And Evidence:**

No

**Requested Changes:**

Requested Changes

I would really like this work to stand the test of time. For Section 4.1, try using llama, or some other publicly available LLM. This could be said of both the zero-shot prediction tasks, as well.

Currently, as of November 27, 2023, many of the Davinci models will be depreciated in early 2024. By the time this is published, will any of this work be reproducible? For example, gpt-3.5-turbo-0613 depreciates on June 13, 2024.

For Section 4.2, I imagine fine-tuning ESM models, which are mentioned in the intro, to be the proper SoA comparison for this task.

In Table 3, column SPP, why is only MLP highlighted, when other methods perform the same?

“The potential reason is that GPT-4 would tend to remove atomic charges in an attempt to improve the octanol-water partition coefficient instead of modifying the hydrophobic fragments into hydrophilic ones. Such a practice could potentially obliterate significant pharmacophores of a drug. Furthermore, GPT-4 often adopts a more conservative approach to modifying the original molecule, primarily excising some fragments to facilitate synthesis, whereas Modof typically opts to append larger new fragments to the molecule.” Evidence? These claims need to be backed up by consistent processing and statistical analysis.

**Strengths And Weaknesses:**

Strengths

For section 4.2, I think it is good that other LLMs that are trained on biological sequences are compared to (AMP-BERT).

I think the insights shown in Table 1 and Table 2, that natural language LLMs can be finetuned and used on protein sequences, is a useful insight.

I think the comparison in Figure 3 of GPT3.5 vs GPT4 is interesting in that it does produce better results, as noted anecdotally by users.

Weaknesses

Throughout the paper, I’m a bit confused by the different usage of GPT4, Davinci-ft, and GPT-3.5. At the time of publication, what is the difference between all of these? Is this information even available to the public?

It is difficult for me to see how reproducible these results are. There is an entire industry of prompt engineering that has sprung up with these models. Have you tried “Let’s think step by step…” To what extent were the prompts modified or updated due to the responses received when developing this approach? Where the prompts never modified? If so, what was the process by which modification occurred? Was their a train/test/validation split?

In Section 4.1, the DNA sequence of the Vacinia virus is used. Is that sequence in the training data? It is a relatively well studied sequence, so this data may be in GPT4 model parameters. Are there other sequences that you can ensure aren’t in the training data?

“Unfortunately, it fails to deliver the longest CDS, incorrectly identifying an ORF with 83 codons instead. When we give the hint by raising the ground truth, it can realize its fault.” This seems anecdotal, and not reproducible.

For Section 4.2, why wasn’t the ESM model used for fine-tuning? It is also an LLM that is mentioned in the intro, trained specifically on protein sequences.

In table 4, it is difficult to determine if the results are significant, as the scales of each of the three columns are different.

---

### Review · Reviewer_y1ch · 2023-11-29

**Summary Of Contributions:**

This paper evaluates several OpenAI GPT models (in prompted and fine-tuned forms) on six bioinformatics tasks. The authors report strong performance of some GPT* models across all tasks.

**Audience:**

Yes

**Broader Impact Concerns:**

I have no ethical concerns.

**Claims And Evidence:**

No

**Requested Changes:**

The following changes would improve my assessment of the manuscript.

**Presentation**
- Add a dataset table summarize the size and source of all datasets
- Discuss other biomedical LLMs (BioGPT, Galactica). There aren't even mentioned in the background literature section.
- Include more data about the tasks themselves (average sequence length)

**Experimental Rigor**
- Evaluate the same set of models across all tasks.
- Use the full test set for all tasks (i.e., no subsampling -- for these scales of test dataset sizes that should be easily doable)
- Include realistic baselines for each task. Start by finding SOTA for each tasks and at least include that.
- Include some non-OpenAI models, e.g., BioGPT as baselines. Only using OpenAI models undermines the work, since OpenAI does not disclose key details on how their models are built.
- Include confidence intervals from test set bootstraps. Are differences meaningful?

Misc Comments/Questions

- AS is the case with all of OpenAI's models, there are no guarantees these datasets aren't included in pretraining. For tasks that are directly available on the web like the ROSALIND dataset or even the AMP-BERT (https://github.com/GIST-CSBL/AMP-BERT/) I have concerns GPT* will have seen these tasks.

- Most of these tasks are short sequence tasks. Some statistics on sequence length would improve clarity and intuition on the difficulty of these tasks.

- The footnote for 2 seems incorrect (points to the moderation model not davinci)

- Under *4.1 Performance on Identifying Potential Coding Regions* The results are more of a qualitative analysis? If the results are literally the 1 example from the figure, that isn't very convincing analysis.

- Under *4.2 Performance on Identifying Antimicrobial Peptide* "Davinci-ft is not a model targeting at proteins" however, you are finetuning the model for protein classification, so this isn't surprising at all. Also not surprising that its capable of beating simpler models like SVMs, MLPs, etc. AMP-BERT is a more realistic baseline.

- Including Table 1 (training set performance of 5-repeated 10-fold CV) isn't very informative. The resulting Davinci-ft model largely underperforms AMP-BERT (Table 2) when evaluated on the test set.

- Task 5's prompt is confusing (or wrong?) How do you use SMILE molecule strings for gene/protein NER?

**Strengths And Weaknesses:**

Strengths

- Evaluating LLMs for tasks outside traditional NLP domain is critical for high-stakes applications of ML
- The authors pick a diverse set of evaluation tasks.

Weaknesses

- The paper's results are not structured clearly. While each task is described, the connection to specific evaluation datasets isn't clear. A main table of all datasets, their sizes, and number of classes would greatly improve the clarity of the paper.

- Experimental methodology is messy: Each evaluation task jumps around to different GPT models: gpt-3.5-turbo-0613, GPT-3.5, fine-tuned davinci, GPT-4. Why not be consistent and evaluate all tasks across the same core set of OpenAI models?

- Baselines are generally weak or straw-man (e.g, SVMs and MLPs). The only existing, strong baseline model for proteins that was evaluated was AMP-BERT. For the NLP/NER tasks the baseline model used is a BiLSTM from 2019, so not state-of-the art by far.

- Missing key references or comparisons with other work (even just for framing the problem). No citation or discussion of models such as Galactica (Taylor et al 2022) which also looks at protein tasks or general biomedical models like BioGPT (Luo et al)

- Key details on the datasets used are missing, e.g., statements such as found in Section 4.4 "A molecule dataset is randomly drawn from the Modof test set" How large of a test set? From the original Chen paper, they used a test set of 800 molecules, but here it looks like only 12 molecules were used.

- Many of the test datasets are very small. Anti-cancer Peptides: 344 instances, Molecule Optimization: 12 instances (?), Educational Problem Solving (105 instances).  I wonder if there is actual statistical power to detect differences here.

- Some measurement of standard error/confidence intervals via test set bootstrapping is needed.

- Findings are overstated given the small scales of the evaluation datasets, e.g.,  "GPT-4 can successfully generate valid SMILES in most cases. It becomes evident that it has assimilated fundamental principles of physical chemistry" This is for a test set containing 12 molecule examples?

---

### Decision · Action_Editor_McRr · 2024-01-11

**Recommendation:** Reject

**Comment:**

Reviewers appreciated that this paper included a wide evaluation of a range of bioinformatics tasks on a range of language models, and felt that it was an important topic to study. However, the issues raised under "claims and evidence" led the reviewer consensus to be to reject the paper. The authors did a great deal of work to clarify the contributions and framing the paper in their rebuttal, but the reviewers felt that the paper still had significant limitations that would need to be addressed before publication.

**Audience:**

Yes.

**Claims And Evidence:**

The paper undertakes an evaluation of various language models (large not not-so-large) on a wide range of bioinformatics tasks. Reviewers raised the following concerns about the accuracy, convincingness, and clarity of the evidence:
1) Different models are used for different tasks, which the reviewers found hard to follow (lack or clarity). The authors partially addressed this concern in the rebuttal.
2) Some of the models being used are closed-source/behind APIs, which makes the results hard to reproduce (and one reviewer tried and failed to reproduce some of the results) given that such models can change over time and/or become deprecated (lack of accuracy).
3) Some of the chosen datasets were sufficiently small that comparison might not be meaningful (lack of convincingness).

**Resubmission Of Major Revision:**

The authors may consider submitting a major revision at a later time.